# Diagnostic Values of Red Flags and a Clinical Prediction Score for Emergent Intracranial Lesions in Non-Traumatic Pediatric Headaches

**DOI:** 10.3390/children9060863

**Published:** 2022-06-10

**Authors:** Atipat Manoyana, Salita Angkurawaranon, Sumintra Katib, Natrujee Wiwattanadittakul, Wachiranun Sirikul, Chaisiri Angkurawaranon

**Affiliations:** 1Department of Radiology, Faculty of Medicine, Chiang Mai University, Chiang Mai 50200, Thailand; atipat_manoyana@hotmail.com (A.M.); sumintratofu@gmail.com (S.K.); 2Global Health and Chronic Conditions Research Group, Chiang Mai University, Chiang Mai 50200, Thailand; chaisiri.a@cmu.ac.th; 3Department of Pediatrics, Faculty of Medicine, Chiang Mai University, Chiang Mai 50200, Thailand; natrujee.w@gmail.com; 4Department of Community Medicine, Faculty of Medicine, Chiang Mai University, Chiang Mai 50200, Thailand; wachiranun.sir@gmail.com; 5Department of Family Medicine, Faculty of Medicine, Chiang Mai University, Chiang Mai 50200, Thailand

**Keywords:** non-traumatic headaches, pediatrics, red-flags, prediction

## Abstract

Introduction: Diagnosis of emergent intracranial lesions that require emergency treatment either medically or surgically in non-traumatic pediatric headaches is important. Red-flag signs and symptoms are commonly used as justification for neuroimaging; however, evidence on its diagnostic values is limited. The study aims to identify diagnostic values of red-flags and develop a clinical prediction score to help improve the diagnostic yield of neuroimaging. Methods: A retrospective review of 109 pediatric patients from 2006 to 2020 who presented with a non-traumatic headache was conducted. A clinical prediction score from red flags was developed using multivariate logistic regression. Discriminatory ability was examined using the area under the receiver operating characteristic curve. Results: A total of 51 patients were diagnosed with emergent intracranial lesions. Four potential clinical red flag predictors were identified: (1) acute onset (less than 3 months), (2) altered conscious state, (3) focal motor abnormality, and (4) and ocular/pupillary abnormality or squint. A clinical prediction score was developed with good discriminatory properties (0.84). Conclusions: Clinical predictor scores from these four red flags may play an important role in maximizing neuroimaging and proper management for pediatric patients with non-traumatic headaches. Future validation studies are needed and could guide referrals and optimize the use of neuroimaging for these patients.

## 1. Introductions

Non-traumatic pediatric headache is a common chief complaint in the pediatric emergency department and in the out-patient department (OPD) [1,2,3]. This is a challenging issue for pediatricians and pediatric neurologists worldwide due to its diagnostic difficulties. Secondary headaches from intracranial lesions, such as tumors, even though rare, are the main concern among these patients. The differentiation of secondary headaches from benign primary headaches is critically important because their managements differ. A patient with intracranial abnormality requires admission for close monitoring, aggressive medical treatment, or emergency surgical consultation, whereas a patient with a primary headache can be treated conservatively as an OPD case.

Emergency neuroimaging (brain CT and MRI), within 24–48 h, has been the investigation of choice to exclude intracranial pathologies and has significant value in guiding the management [4]. However, there are some precautions regarding neuroimaging, which includes economic issue [2,5], radiation exposure [6], contrast media allergy, contrast-induced nephropathy [7], and risks from sedative drugs [8]. Clinical practice depends greatly on the clinical predictors, the so-called “red-flags”, to predict the likelihood of intracranial pathologies [3,7,9]. 

However, some red-flags, such as fever, are non-specific for intracranial lesions. Other red-flags, such as headache locations, the characteristics of the headache, and headaches aggravated by the Valsalva maneuver, can be unreliable due to potential communication issues in the pediatric population. Recent evidence, published in 2019, suggests that many children with headaches may be receiving unnecessary neuroimaging due to the high prevalence of non-specific red flag signs and low prevalence of clinically significant intracranial lesions [10].

In Thailand, the decision for imaging referral and level of emergency depends primarily on clinical judgment in individual cases. For primary care centers in Thailand, the differentiation of secondary headaches from primary ones by clinical information is crucial as they lack sophisticated equipment and easy access to neuroimaging [11].

While the presence of these red-flag signs is associated with intracranial pathologies, in order to optimize the use of neuroimaging, the study aims to evaluate the predictive values of each red-flag and develop a concise clinical prediction score from red-flags to help identify those children with non-traumatic headaches who are at risk of having an emergent intracranial lesion.

## 2. Methods

A retrospective review of electronic health records of all the pediatric patients (defined as age 15 or under in Thailand [12]) was conducted from January 2006 to December 2020 at Chiang Mai University Hospital. Chiang Mai University Hospital is the largest university referral Hospital in Northern Thailand. The only initial inclusion criteria were children who presented with a chief complaint of headache and underwent neuroimaging assessment during the period of review; however, our main exclusion criteria were those with a history of trauma, absence of clinical data, or poor image quality. A total of 208 patients were reviewed. 

Of these, 94 patients were excluded due to a history of trauma, four patients were excluded due to absent clinical data, and one patient was excluded due to poor image quality. Finally, 109 patients were included in the study (Figure 1). Clinical data for red flag signs/symptoms were reviewed by a pediatrician (NW) and a family physician (CA), who were blinded from the neuroimaging findings, while neuroimaging findings (from Brain CT, CTA, and MRI images) were reviewed by a radiologist (AM) and a neuroradiologist (SA) blinded from the red-flag findings. From 109 patients, 112 scans were obtained from three different modalities (87 CT scans, one Brain CTA, and 24 MRI scans). Any disagreement between the reviewers was resolved by consensus.

### 2.1. Red-Flag Clinical Predictors

The variable was classified as the presence and absence of 23 red-flags (Appendix A), which was adapted from the recently published criteria by Raucci et al. in 2019 [3]. These include severe vomiting, fever, focal motor abnormality, visual field deficiency, abnormal ocular movement, and seizures. In our study, we added “acute onset” (of less than three months) as another red-flag from the criteria by Raucci et al. This was because acute or sudden onset headache was considered as one of the red-flags in previously proposed criteria for both children and adults [13,14,15].

### 2.2. Emergent Intracranial Lesions

The definition for emergent intracranial lesions was the presence of intracranial lesions from neuroimaging that required emergency treatment either medically or surgically or admission for close monitoring, and if left untreated, the patient would rapidly deteriorate. By this definition, examples of positive results include large intracranial hemorrhage requiring surgical removal, ruptured intracranial aneurysm requiring surgical clipping, acute cerebral venous sinus thrombosis requiring medical thrombolysis, and brain abscess requiring antibiotic treatment. Negative results would include cavum septum pellucidum that does not require further treatment, benign lesions, such as small intracranial calcified granuloma or arachnoid cysts that do not require further management and incidental findings, such as a pineal cyst. A full list of conditions is provided as Appendix A.

### 2.3. Statistical Analyses

Descriptive statistics were used to describe the sample demographics, the prevalence of each red flag sign, and the final diagnosis. The association between the presence of each red flag and emergent intracranial lesions was tested using chi-square if the expected value for each cell was five or greater. Otherwise, Fischer’s exact was used. Multivariable logistic regression was used to derive the clinical prediction score. All candidate red flags signs significant in the univariable analysis were entered into the model (*p* < 0.10). At this stage in variable selection, using conventional *p*-value cut-offs (*p* < 0.05) may result in important predictors being excluded [16]. A backward stepwise approach was then used to reduce red-flag predictors to produce a final model. The backward stepwise approach is often the preference for stepwise selection as it starts with the full model [17]. 

As supported by the literature, the significance level for removal was set at 0.15, and the significance level for addition was set at 0.10 [18]. Dividing by the smallest log-odds coefficient in the final model and rounded to the nearest integer, weights were assigned to each coefficient. The final diagnostic score was derived from the sum of the weighted score of each red flag sign. The sensitivity, specificity, positive likelihood ratio, and negative likelihood ratio of the prediction score were calculated to examine the diagnostic accuracy of the score. The discriminatory ability was quantified using the area under the receiver operating characteristic curve (AuROC). The Hosmer–Lemeshow goodness of fit test was used to determine the model fit. All analyses were performed using STATA version 15.

## 3. Results

The mean age of the 109 pediatric patients with non-traumatic headaches was 10.6 (SD 3.4), and 46 (42%) were male. Only 5% of patients were younger than the age of five. The majority (80.7%) received a CT scan (80.7%), while 22% had an MRI performed. Two patients (1.8%) underwent both CT scan and MRI during the same event. The presence of comorbidities was present among 26 patients (23.8%). Common comorbidities include hematologic malignancy, systemic lupus erythematosus, HIV/AIDS, congenital heart disease, and thalassemia. Fifty-one patients (46.8%) were identified as having an emergent intracranial lesion. However, there were no significant differences in demographic characteristics between those with and without emergent intracranial lesions (Table 1).

Table 2 describes common final diagnoses among those with and without emergent intracranial lesions. The three most common final diagnoses among those with emergent intracranial lesions (*n* = 51) were brain tumors (25.5%), intracranial infections (25.5 and intracranial hemorrhage (13.7%). In non-emergent intracranial lesions (*n* = 59), the three most common final diagnosis were migraine (34.5%), inconclusive diagnosis (24.1%), and tension-type headache (10.3%). The patients who were classified as “inconclusive diagnosis” were those with unremarkable imaging findings and had no definitive cause of headache identified upon further investigations. These patients were treated conservatively. A detailed breakdown of specific diagnoses can be found in Appendix A.

Acute onset, severe vomiting, and high-risk comorbidities were the three most common red-flags found among 72 patients (66.1%), 65 patients (59.6%), and 26 (23.8%), respectively (Table 3). Table 3 reports their association with emergent intracranial lesions. In the crude analysis, the red-flags that showed statistically significant relationships with emergent intracranial lesions include acute onset, focal motor abnormality, changes in mood or personality, altered conscious state, seizures, abnormal ocular movements, ataxia, and meningism. 

Detailed analysis and predictive value of each of the 23 red flags can be found in Appendix A. However, from multivariable analyses, only four red flags demonstrated potential as significant predictors of emergent intracranial lesions: 1: acute onset (OR 5.24, 95% CI 1.60 to 17.1), 2: altered conscious state (OR 3.07, 95% CI 0.80 to 11.70), 3: focal motor abnormality (OR 10.06, 95% CI 2.32 to 43.22) and 4: abnormal ocular/pupillary movements (OR 19.87, 95% 3.54 to 111.6). The model fitted the data reasonably well (Hosmer–Lemeshow Chi-square = 2.34, *p* = 0.67). Based on the coefficients, these variables were assigned weights of one, one, two, and three, respectively. (Table 4).

The proposed clinical predictor score, which now ranges from 0 to 7, showed good discriminatory properties with an area under the receiver operating characteristic curve (AuROC) of 0.884 (95% CI 0.775 to 0.914) (Figure 2). Using a score of 2 as the cut-off for high risk of emergent intracranial lesions correctly identified 78% of cases, with a sensitivity of 68.6% and specificity of 86.2%. A score of at least 2 would increase the likelihood of having an emergent intracranial lesion by nearly five folds (LR + 4.98). A cut-off score of 3, while demonstrating lower sensitivity (51%), would provide greater specificity (93%) and a positive likelihood ratio of 7.39 (Table 5).

## 4. Discussion

While it is known that red flag signs and symptoms are associated with emergent intracranial lesions, their diagnostic values, especially for non-traumatic pediatric headaches, remain underexamined. The study provides detailed diagnostic and predictive values of red flag signs and symptoms. In addition, the study provides a clinical prediction scoring system based on four red flags (acute onset, altered conscious state, focal motor abnormality, and abnormal ocular/pupillary movements), which demonstrated good discriminatory properties. 

The prevalence of emergent intracranial lesions in this study was 46.7%, much higher than the 4% prevalence of space-occupying lesions in childhood headaches that underwent imaging from a study by Medina et al. from the United States [19]. The higher incidence of a positive outcome in our study is likely due to the nature of our institution as a referral center for diagnostics and treatment. It is important to still note that the majority of intracranial lesions were brain tumors, infection, and intracranial hemorrhage, which was quite similar to the prevalence from the previous studies [19,20,21,22]. For non-emergent intracranial lesions, the most common diagnosis was a primary headache (23.9%), such as migraines, which was also similar to previous reports [19,20,21,22].

The lack of a clear consensus guideline on indicators for neuroimaging in Thailand was reflected in our study, where nearly 45% of patients with non-emergent lesions were diagnosed with primary headaches. The study of Rho et al. [23] also found that the rate of unnecessary imaging was high in pediatric headaches, and a large portion of patients undergoing imaging were those with recurrent headaches. Only 9.3% had abnormal findings on neuroimaging, and 0.71% required surgical excision. The authors also highlighted the need for rational guidelines for neuroimaging in pediatric headaches.

Of the 23 red-flags explored in our study, four had promising predictive values, which included ocular motor/pupillary abnormality or squint, altered conscious state, focal motor abnormality, and the acute onset of headache. Acute onset of headache (headache of fewer than three months) was statistically related to significant intracranial lesions in our study. This is likely because common intracranial lesions that require close medical attention, such as intracranial hemorrhage, cerebral infection, and stroke, usually present in an acute form. On the other hand, chronic recurrent headaches tend to be associated with primary headaches, such as migraine or tension-type headaches. The American Academy of Neurology and the Child Neurology Society guidelines also suggested that neuroimaging is not indicated in children with recurrent headaches and a normal neurologic examination [24].

Focal motor abnormality was a significant sign according to our study. This is similar to previous studies [19,23]. The existence of space-occupying lesions, such as a tumor, bleeding, brain abscess, and infarction, often cause a motor deficit, especially when located adjacent to the corticospinal tract. The advantages of this sign are that it is simple and reliable. 

Motor power examinations can also help differentiate hemiplegic migraine from true intracranial lesion because the motor function is usually preserved in hemiplegic migraines. Ocular movement/pupillary abnormality also had a great predictive value with an odds ratio of 19.9 in our study. The literature supports that it is one of the most commonly found signs in cases of serious neurologic conditions [25]. Unlike the more invasive eye-ground examination for papilledema, the ocular/pupillary examination can be considered superior in aspects of its noninvasiveness, readily available without requiring sophisticated equipment, and can be performed in nearly all age groups. Our last prediction of the four red-flags predictors, alteration of the mental status, had the lowest odds ratio. This is concordant with the clinical situation that other extracranial causes, such as sepsis, hypovolemia, and electrolyte imbalance, may also induce mental status changes in children [26]. 

A number of red flags were not significantly associated with emergent intracranial lesions in our study. Some signs and symptoms can be found in many different conditions and thus, are less specific. Severe vomiting, for example, may occur in patients with migraine or post-chemotherapy. Headache with fever can be found in a wide variety of systemic diseases and infections, which is much more common than intracranial pathology. Some red flags, such as changes in mood or personality, are quite subjective and depend greatly on the perception of the parents and the attentiveness of their children’s monitoring. 

The occipital location of the headache was also not statically significant in our study. The recently updated red flags criteria by Raucci et al. considered the occipital headache as a “relative” red flag [3]. Moreover, some studies even proposed that the location of the headache may not be correlated with a significant intracranial lesion [27]. Lastly, other red-flag signs, such as worsening pain with cough or Valsalva maneuver, headache character, and headache location, in young children are potentially difficult due to communication boundaries. Thus, these red-flags may be less specific and have a low interobserver agreement, resulting in lower diagnostic values.

In our study, only six patients were younger than five years of age. The cut-off point of 5 years did not reach statistical significance, likely from the small number of patients in this age group. However, we found that most of them were diagnosed with brain tumors, such as ependymoma, medulloblastoma, and pilocytic astrocytoma. This suggests that emergent intracranial lesion is still a matter of concern for patients presenting with non-traumatic headaches at an extremely young age. 

One red-flag that did not exist in our patient population was “change in the character/pattern of headache in patients previously diagnosed with a primary headache”. One study found that “change in the type of headache” demonstrated low predictive yield in children with recurrent headaches, and the author even suggested removing “change in the type of headache” from the official guidelines for children [23]. 

In non-traumatic adult headaches, clinical prediction scores have proven to be beneficial in guiding high-risk patients for neuroimaging with an ability to exclude serious intracranial pathologies [28,29]. Our study developed a similar clinical prediction score for non-traumatic headaches in pediatrics using four red-flags. The score ranged from 0 to 7 with an AuROC of 0.84, demonstrating good discriminatory and diagnostic probabilities. A validated clinical prediction score could serve as a rational, quantitative and evidence-based tool on the basis of available history taking and physical examination to facilitate the clinician’s referral judgment [30]. 

This situation is of importance in the context of Thailand and other low-to-middle-income countries, where the ratio of CT or MRI units per population is rather suboptimal [31]. As radiological investigations are capital- and labor-intensive, the CT or MRI units are available only at some secondary or tertiary care centers in these countries. The capability in diagnosis and treatment depends greatly on the referral system and requires locally generated evidence on patterns and the likelihood of emergent intracranial lesions.

The clinical score can help primary care physicians select the proper patient to be referred to the higher center for specialist consultation and further neuroimaging [32]. On the other hand, in the context of tertiary care centers, the score assists emergency physicians, pediatricians, and pediatric neurologists in classifying headache patients according to their likelihood of having an intracranial lesion. The higher score urges the clinician to seek emergency neuroimaging and intensive management. The implementation of the score also reduces the variability in management among clinicians, moreover, optimizes the use of neuroimaging resources, and maximizes the predictive value of brain scans. 

In our study, a cut-off of 2 had high specificity (86.2%) and was able to classify most patients without emergent lesions correctly. This cut-off point also correctly classified 78% of all patients, was able to identify nearly 70% of positive cases, and yielded a positive likelihood ratio of almost 5. A cut-off point of 3 had higher specificity (93.1) but had low sensitivity (51%). Given the potential severity of undetected emergent lesions, the cut-off point of 2 could be a useful starting point for future studies in similar settings. 

There are several limitations to this study. As the study had a retrospective design, and all signs and symptoms were reviewed from recorded data. Some signs may be under-evaluated, thus, limiting its diagnostic potential. The study was conducted from a single study center in Thailand, which could also limit its generalizability. With no clear consensus guideline for neuroimaging in non-traumatic pediatric headaches in Thailand, coupled with limited accessibility of neuroimaging and pediatric neurologists to only large referral centers; this resulted in a small pool of patients over the 15-year period. Further multicenter prospective studies would be valuable in validating the diagnostic values of red-flags and improving the efficacy of the clinical predictor scores.

## 5. Conclusions

Pediatric patients are more fragile to under- and over-investigation. Among non-traumatic pediatric headaches, there is a lesser amount of research information concerning clinical predictors of emergent intracranial lesions. Our study suggests that clinical predictor scores from four red flags: (1) acute onset (less than 3 months), (2) altered conscious state, (3) focal motor abnormality, and (4) and ocular/pupillary abnormality or squint may play important roles in maximizing neuroimaging and proper management for those patients in Thailand. Future validation studies are needed and could guide referrals and optimize the use of neuroimaging for pediatric patients with non-traumatic headaches in Thailand and other countries where the burden and distribution of causes of non-traumatic headaches may be similar.

## Figures and Tables

**Figure 1 children-09-00863-f001:**
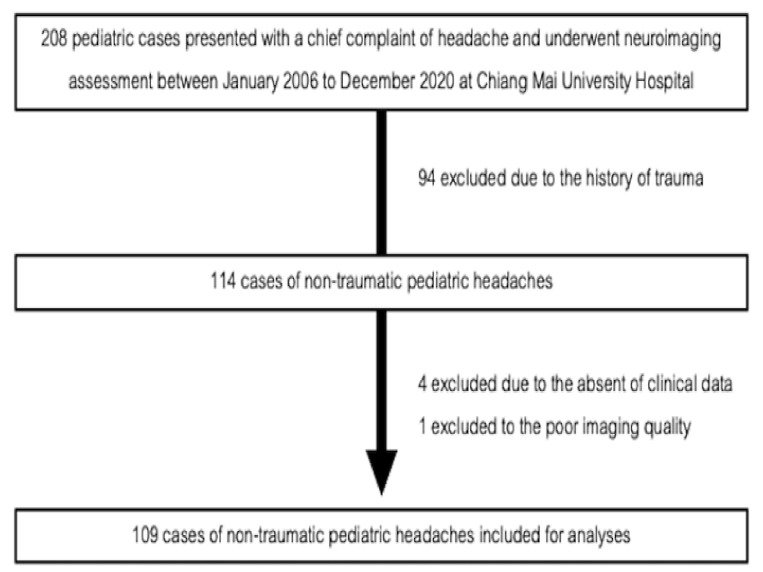
Flowchart of patient selection.

**Figure 2 children-09-00863-f002:**
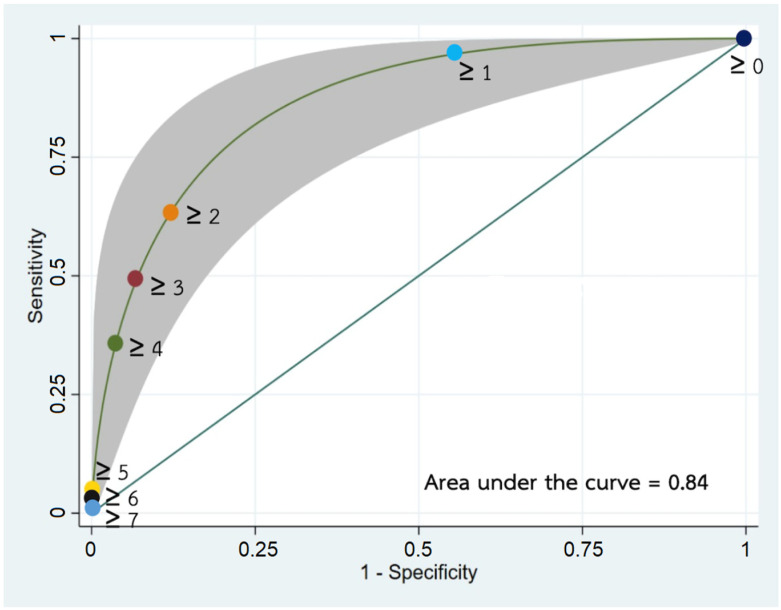
Area under the receiver operating characteristic curve (AuROC) of the proposed clinical predictor score.

**Table 1 children-09-00863-t001:** Baseline demographics.

Demographics	Total (*n* = 109)	Emergent Intracranial Lesion		
		Negative Lesions (*n* = 58)	Positive Lesion (*n* = 51)	*p*-Value
Mean Age (SD)	10.6 (3.4)	11.2 (3.1)	10.1 (3.7)	0.09
Age < 5 (*n*, col %)	6 (5.5)	2 (3.5)	4 (7.8)	0.32
Male (*n*, col %)	46 (42.2)	27 (46.6)	19 (37.3)	0.33
Presence of high-risk comorbidities; (hematologic malignancy, SLE, and HIV infection).	26 (23.8)	11 (19.0)	15 (29.4)	0.20

**Table 2 children-09-00863-t002:** The final diagnosis of the pediatric patients with non-traumatic headaches.

Diagnosis	Frequency	Percentage
**Emergent intracranial lesion (*n* = 51)**		
Brain tumor	13	25.5%
Intracranial infection	13	25.5%
Intracranial hemorrhage	7	13.7%
**Non-emergent intracranial lesion (*n* = 58)**		
Headache disorders	26	44.8%
Migraine	20	34.5%
Tension-type headache	6	10.3%
Inconclusive diagnosis	14	24.8%
Others, such as dengue fever, epilepsy, chronic sinusitis, and hypertension	11	19.0%

**Table 3 children-09-00863-t003:** Association between red flags and emergent intracranial lesions.

Common Red-Flags (*n*, col %)	Total (*n* = 109)	Emergent Intracranial Lesion		
		Negative Lesions (*n* = 58)	Positive Lesion (*n* = 51)	*p*-Value
Acute onset (<3 months)	72 (66.1)	30 (51.7)	42 (82.4)	<0.01
Severe vomiting	65 (59.6)	31 (53.5)	34 (66.7)	0.16
High-risk underlying comorbidities	26 (23.8)	11 (19.0)	15 (29.4)	0.20
Fever	21 (19.3)	8 (13.8)	13 (25.5)	0.12
Focal motor abnormality	20 (18.4)	3 (5.2)	17 (33.3)	<0.01
Changes in mood or personality over days or weeks	20 (18.4)	5 (8.6)	15 (29.4)	<0.01
Altered conscious state	19 (17.4)	4 (6.9)	15 (29.4)	<0.01
Seizures	16 (14.7)	5 (8.6)	11 (21.6)	0.06
Abnormal ocular movements, squint, pathological pupillary responses	16 (14.7)	2 (3.5)	14 (27.5)	<0.01
Increase in severity or characteristics of the headache	14 (12.8)	5 (8.6)	9 (17.7)	0.16
Pain that wakes the child from sleep or occurs on waking	13 (11.9)	8 (13.8)	5 (9.8)	0.52
Ataxia, gait abnormalities, impaired coordination	13 (11.9)	1 (1.7)	12 (23.4)	<0.01
Meningism	12 (11.0)	3 (3.5)	10 (19.6)	<0.01
Occipital headache	11 (10.1)	8 (13.8)	3 (5.9)	0.17

**Table 4 children-09-00863-t004:** Clinical Predictors for Emergent Intracranial Lesions in Children with non-traumatic Headaches.

Red-Flags	Odds Ratio (95% CI)	*p*-Value	Coefficient	Weight
Acute onset (<3 months)	5.24 (1.60 to 17.1)	<0.01	1.656654	1
Altered conscious state	3.07(0.80 to 11.79)	0.10	1.120896	1
Focal motor abnormality	10.06 (2.34 to 43.22	<0.01	2.308371	2
Abnormal ocular movements, squint, pathologic pupillary responses	19.87 (3.54 to 111.58)	<0.01	2.989329	3

**Table 5 children-09-00863-t005:** Diagnostic accuracy of the clinical predictor score.

Cut-Point	Sensitivity	Specificity	Correctly Classified	Positive Likelihood Ratio	Negative Likelihood Ratio
≥1	96.1%	44.8%	68.8%	1.74	0.08
≥2	68.6%	86.2%	78.0%	4.98	0.36
≥3	51.0%	93.1%	73.4%	7.39	0.53
≥4	33.3%	98.3%	67.9%	19.3	0.68
≥5	5.9%	98.3%	55.1%	3.41	0.96
≥6	3.9%	100%	55.5%		0.96
≥7	2.0%	100%	54.1%		0.98

AuROC 0.844 (95% CI 0.775 to 0.914).

## Data Availability

Data cannot be shared publicly because it was not formally approved by the Ethics Committee. Anonymized Data are available from the Faculty of Medicine, Chiang Mai University Research Administration Office (contact via researchmed@cmu.ac.th) for researchers who meet the criteria for access to confidential data.

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
