# Peer review of "Diagnostic Values of Red Flags and a Clinical Prediction Score for Emergent Intracranial Lesions in Non-Traumatic Pediatric Headaches"

_children, 2022, doi:10.3390/children9060863_

Round 1

Reviewer 1 Report

This study aimed to evaluate the value of clinical red flags in predicting emergent intracranial lesion on neuroimaging.

Abstract:

  • Why 109 patients? Did you perform sample size calculations?
  • Please specify your study design, time period of inclusion and inclusion and exclusion criteria in your abstract
  • “with emergent intracranial lesions” – what was the definition of emergent lesion? Please explain briefly in the abstract.
  • “in maximizing neuroimaging” – please rephrase, this is somewhat misleading
  • “headache in Thailand” – why only in Thailand?

Introduction

  • “Emergency neuroimaging (brain CT and MRI)” – what timeframe do you consider emergency imaging? Please define.

Methods

  • “all the pediatric patients ( age 15 under)” – why a cut-off age of 15 years?
  • Please start the methods section with a clear IRB statement. Even for a retrospective study, you need an IRB approval/ waiver.
  • Please provide clear inclusion and exclusion criteria.
  • “reviewed by clinicians” – what kind of “clinicians”?

Results

  • The rate of emergent lesions is unusually high – I wonder why it is that high? In most settings it would be much lower.
  • Also, the number pf patients included that were 5 years or younger seems unusually small

Discussion

  • “pediatric headache is a specific entity, which results in a small pool of patients over the 15-year period” – in our experience, pediatric headache is very common.

Reviewer 2 Report

Introduction

- cited studies provide appropriate context for current work. Could provide more context for conducting the study in Thailand, as is described in the discussion section of the paper.

Methods

- need to justify why the age cutoff of 15 and younger

- include a flow chart showing inclusion and exclusion criteria and associated 'n' values as described in text. Be more explicit about inclusion and exclusion criteria.

- Good use of blinding with respect to red-flag and imaging analyses by clinicians

- Under 'red-flag clinical predictors', cite Table S3 as it contains the complete criteria used in your study and the prevalence within your sample

- Adding 'acute onset' of headache as an additional criteria is appropriate

- Under 'emergent intracranial lesions', please be more comprehensive about what is considered 'emergent' and 'non-emergent'. You currently list a few options but it is important for reproducibility and further testing and comparisons of your model, to include detailed criteria. If the list is not too long, it can be placed in the supplementary data.

- Under 'statistical analyses', it is important to clearly describe if you performed chi-square or Fisher's exact test instead of saying both. GraphPad has a helpful guide describing the two tests: GraphPad Prism 9 Statistics Guide - Fisher's test or chi-square test? 

- Please describe what your multivariable model controls for such as age, sex, etc.

- Please clarify what the 'crude analysis' refers to in the methods and why you chose p<0.1 as your cutoff.

- Please also explain why you decided to use a backward stepwise regression instead of forward stepwise regression. Please describe your process for removing variables from the model (ie. p-value threshold, RSS threshold, adjusted R2 threshold, AIC/BIC, etc.) and how you determined a threshold to stop removing variables from the model.

- Do you have any measures of model fit that can also be included with the results such as AIC?

Results

- Table 1 has formatting issues with brackets

- Table 4 shows 'altered conscious state' as having a p-value of 0.1 while the others are 0.01 or lower. Please indicate is this is a mistake and, if it is not, why this variable was included in the model despite a p-value of 0.1.

- ROC figure and associated table are well done

Discussion

- The discussion has information that could be moved to the introduction. For example, the discussion of neuroimaging constraints in Thailand could be moved to the introduction ('In Thailand....')

- There could be more discussion of deciding on a cutoff score and the associated sensitivity, specificity, and LRs. I understand there is a small sample size but it would be helpful to understand what the author's thoughts are surrounding the tradeoff of sensitivity and specificity in relation to clinical decision making with this new tool.

Conclusion

- Sentence fragment at the beginning of the paragraph

- Elaborate on what the four red flags are to solidify the conclusion

Round 2

Reviewer 1 Report

The authors have responded to my previous comments and questions. I have no further comments. 

Reviewer 2 Report

Excellent job addressing comments from myself and the other reviewer. After reading through the comments from the other reviewer and your edits, I believe you addressed our concerns in a satisfactory manner and were transparent about the shortcomings of your study. Your phrasing of your model as a starting point for broader studies, and not jumping to broad conclusions straight away, is well received.